# Regulation of plant phototropic growth by NPH3/RPT2-like substrate phosphorylation and 14-3-3 binding

Stuart Sullivan [1✉], Thomas Waksman [1], Dimitra Paliogianni [1], Louise Henderson[1], Melanie Lütkemeyer[1,2], Noriyuki Suetsugu[1,3] & John M. Christie [1✉]

Polarity underlies all directional growth responses in plants including growth towards the light (phototropism). The plasma-membrane associated protein, NON-PHOTOTROPIC HYPO-COTYL 3 (NPH3) is a key determinant of phototropic growth which is regulated by phototropin (phot) AGC kinases. Here we demonstrate that NPH3 is directly phosphorylated by phot1 within a conserved C-terminal consensus sequence (RxS) that is necessary to promote phototropism and petiole positioning in *Arabidopsis*. RxS phosphorylation also triggers 14-3-3 binding combined with changes in NPH3 phosphorylation and localisation status. Mutants of NPH3 that are unable to bind or constitutively bind 14-3-3 s show compromised functionality consistent with a model where phototropic curvature is established by signalling outputs arising from a gradient of NPH3 RxS phosphorylation across the stem. Our findings therefore establish that NPH3/RPT2-Like (NRL) proteins are phosphorylation targets for plant AGC kinases. Moreover, RxS phosphorylation is conserved in other members of the NRL family, suggesting a common mechanism of regulating plant growth to the prevailing light environment.

[1] Institute of Molecular, Cell and Systems Biology, College of Medical, Veterinary and Life Sciences, Bower Building, University of Glasgow, Glasgow G12 8QQ, UK. [2] Present address: RNA Biology and Molecular Physiology, Faculty of Biology, Bielefeld University, 33615 Bielefeld, Germany. [3] Present address: Graduate School of Arts and Sciences, The University of Tokyo, Tokyo 153-8902, Japan. ✉email: Stuart.Sullivan@glasgow.ac.uk; John.Christie@glasgow.ac.uk

The ability to sense and respond to the prevailing light conditions is instrumental for plants to adapt their growth and development to the external environment. Phototropism allows plants to re-orientate shoot growth towards a directional light source, which promotes light capture and early seedling growth[1]. Phototropism is induced by UV/blue light and is mediated by two phototropin (phot) light-activated kinases, phot1 and phot2[2]. Phot1 is the primary phototropic receptor and functions over a wide range of fluence rates, whereas phot2 activity requires higher light intensities[3]. Phots also control physiological responses such as chloroplast movement, leaf positioning, leaf expansion and stomatal opening[4], which together serve to optimise photosynthetic efficiency and growth[5–7].

Phototropins are plasma membrane-associated kinases containing two light, oxygen, or voltage-sensing domains (LOV1 and LOV2) at their N-terminus, which bind oxidised flavin mononucleotide (FMN) as a UV/blue light-absorbing cofactor[8,9]. Light perception, primarily by LOV2, results in activation of phototropin kinase activity and receptor autophosphorylation[10,11]. Although multiple phosphorylation sites have been identified within phot1 and phot2[12], only sites within the kinase activation loop have been shown to be important for signalling, and kinase-inactive variants of phot1 and phot2 are non-functional[13,14]. Despite the importance of phot kinase activity for downstream signalling, only a limited number of substrates have been identified to date. BLUE LIGHT SIGNALLING 1 (BLUS1) and CONVERGENCE OF BLUE LIGHT AND CO$_2$ 1 (CBC1) are phot1-kinase substrates involved in blue light-induced stomatal opening[15,16], while phosphorylation of ATP-BINDING CASSETTE B19 (ABCB19) and PHYTOCHROME KINASE SUBSTRATE 4 (PKS4) by phot1 modulates hypocotyl phototropism[17–19]. Given the variety of physiological responses mediated by phot signalling, further phot kinase substrates likely await identification[20].

Phototropism results from the establishment of lateral gradients of the phytohormone auxin, which leads to increased cell expansion on the shaded side of the hypocotyl[1]. NON-PHOTOTROPIC HYPOCOTYL 3 (NPH3) is an essential signalling component for phototropism and is required for the formation of the lateral auxin gradients[21,22]. NPH3, together with

ROOT PHOTOTROPISM 2 (RPT2), are the founding members of the NPH3/RPT2-Like (NRL) protein family, which contains 33 members in *Arabidopsis*[23,24]. The primary amino-acid structure of NPH3 can be separated into three regions based on sequence conservation with other NRL proteins: an N-terminal BTB (bric-a-brac, tramtrack and broad complex) domain, a central NPH3 domain and a C-terminal coiled-coil domain[24]. The C-terminal portion of NPH3, including the coiled-coil domain, is proposed to facilitate localisation of NPH3 to the plasma membrane[25] as well as mediating direct interaction with phot1[21]. NPH3 is reported to function as a substrate adaptor in a CULLIN3-based E3 ubiquitin ligase complex targeting phot1 for ubiquitination[26]. Ubiquitination of phot1 may be involved in receptor desensitisation, particularly under high-light irradiation[26], but its importance in phot1 signalling is currently unknown.

Although the biochemical function of NPH3 remains unresolved, activation of phot1 by blue light results in dynamic changes to NPH3 phosphorylation status and subcellular localisation[27,28]. NPH3 is phosphorylated on multiple sites in darkness, including sites located towards the N-terminus within the NPH3 domain[29], and localises to the plasma membrane[27]. Upon blue light perception, NPH3 is rapidly dephosphorylated[30] and becomes internalised into aggregates, which transiently attenuates its interaction with phot1[27,28]. These effects are reversible in darkness, with the kinetics of NPH3 rephosphorylation matching the photoactive lifetime of phot1[7]. The kinases and phosphatases that modulate NPH3 phosphorylation status are unknown, however, reduced levels of dephosphorylation, and relocalisation into aggregates, correlates with enhanced phototropic responsiveness observed in de-etiolated (green) seedlings[28].

Along with NPH3, two other NRL family members also have known roles in phot signalling pathways. RPT2 interacts with both phot1 and NPH3[31,32], it is proposed to influence NPH3 phosphorylation status and promote the reconstitution of the phot1–NPH3 complex to sustain signalling under higher light intensities[27]. In line with this, phototropic responsiveness in mutant seedlings lacking RPT2 decreases as light intensity is increased[33]. Similarly, *RPT2* expression levels are low in darkness, but increase with irradiation in a fluence-dependent manner[33]. RPT2, together with NPH3, is also involved in phot-mediated leaf positioning and leaf expansion responses[25,34]. NRL PROTEIN FOR CHLOROPLAST MOVEMENT 1 (NCH1) is positioned within the same clade as RPT2 in the *Arabidopsis* NRL phylogenetic tree[24]. NCH1 and RPT2 redundantly mediate chloroplast accumulation movements in response to low-intensity light[35].

Phot signalling is dependent upon reversible changes in phosphorylation[12]. 14-3-3 proteins are present in all eukaryotic organisms and bind to target proteins through the identification of phospho-serine/threonine motifs[36,37]. 14-3-3 binding can produce a variety of consequences, such as regulation of enzymatic activity, changes in subcellular localisation, protein stability or alteration of protein–protein interactions[38]. 14-3-3 proteins are known to bind to phot1 and phot2 following receptor autophosphorylation[13,32,39,40], whereas NPH3 and RPT2 have both been identified as components of the 14-3-3 interactome[41,42]. However, the functional relevance of these interactions and the roles of 14-3-3 proteins in phot signalling remains unclear.

Despite the importance of NRL proteins in blue light-mediated responses, how signalling is initiated upon phot activation is still not known. In the present study, we identify NPH3 as a substrate for phot1-kinase activity. Phosphorylation of NPH3 at the C-terminus by phot1 results in 14-3-3 binding, which is required for early signalling events and promotes NPH3 functionality. The C-terminal phosphorylation site of NPH3 is conserved in several

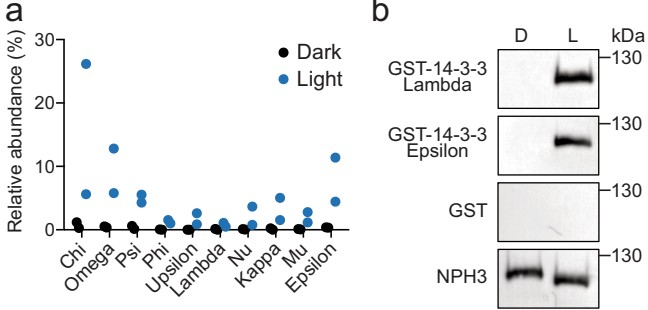

**Fig. 1 NPH3 interacts with 14-3-3 proteins in a light-dependent manner. a** NPH3-interacting proteins were identified by mass spectrometry analysis of anti-GFP immunoprecipitations from etiolated seedlings expressing GFP-NPH3 maintained in darkness (dark) or irradiated with 20 μmol m$^{-2}$ s$^{-1}$ of blue light for 15 min (Light). Protein signal intensities were converted to the relative abundance of the bait protein (GFP-NPH3). Values from two independent biologicals replicates are shown. **b** Far-western blot analysis of anti-GFP immunoprecipitations from etiolated seedlings expressing GFP-NPH3 maintained in darkness (D) or irradiated with 20 μmol m$^{-2}$ s$^{-1}$ of blue light for 15 min (L). GST-tagged 14-3-3 isoforms (Lambda and Epsilon) or GST alone were used as probes. Blots were probed with anti-GFP antibodies as a loading control (bottom panel).

NRL family members, including RPT2, suggesting phot-mediated phosphorylation and 14-3-3 binding may represent a conserved mechanism of regulation.

## Results

**Light-dependent 14-3-3 binding to NPH3.** In order to identify additional components involved in blue light signalling, GFP-NPH3 was immunoprecipitated from etiolated *nph3* mutant seedlings expressing functional *NPH3::GFP-NPH3*[28]. Anti-GFP immunoprecipitations (IPs) were performed on total protein extracts from seedlings maintained in darkness or after a brief blue light treatment (20 μmol m$^{-2}$ s$^{-1}$ for 15 min) to capture early signalling events. Co-purifying proteins were analysed by label-free quantitative tandem mass spectrometry (MS; Supplementary Data 1) to allow the identification of proteins whose abundance changed following blue light irradiation. As expected, phot1 was recovered in the IPs from both dark- and light-treated seedlings, but at a higher abundance in the dark (Supplementary Data 2). This is in agreement with previous results showing NPH3–phot1 interactions are attenuated by blue light[27]. Conversely, several 14-3-3 isoforms were detected at greater abundance following blue light irradiation (Fig. 1a).

14-3-3 proteins bind to target proteins through recognition of phospho-serine/threonine-containing motifs. *Arabidopsis* expresses 13 different 14-3-3 isoforms that can be phylogenetically divided into the epsilon and non-epsilon groups[43]. Far-western blotting was performed to assess direct 14-3-3 binding to GFP-NPH3. Binding of recombinant 14-3-3 Lambda (non-epsilon group member) and 14-3-3 Epsilon (epsilon group member) fused to glutathione-S-transferase (GST) was not detected for GFP-NPH3 IPs from etiolated seedlings maintained in darkness (Fig. 1b). Blue light irradiation results in enhanced electrophoretic mobility of GFP-NPH3 owing to its rapid dephosphorylation[30]. Concurrently, binding of 14-3-3 Lambda and Epsilon was observed following irradiation, whereas no binding was observed when GST alone was used as the probe. In line with the results from IP-MS analysis, no specificity in binding of 14-3-3 proteins from epsilon and non-epsilon groups was detected. These results suggest that blue light irradiation triggers both phosphorylation, and concomitant 14-3-3 binding, as well as dephosphorylation events on NPH3.

**Analysis of phosphorylation sites within NPH3.** Activation of phot1 by blue light results not only in rapid changes in the phosphorylation status of NPH3 but also its subcellular localisation[27,28]. In the darkness, NPH3 localises predominantly to the plasma membrane but is rapidly internalised into aggregates upon blue light treatment. Based on data from global phosphoproteomics experiments[44,45] three regions of NPH3 (M1, M2 and M3) containing the majority of experimentally identified phosphopeptides were selected for mutational analysis (Fig. 2a). Within each of the regions, all of the serine and threonine residues were replaced with alanine to mimic the dephosphorylated state. The mutations were introduced into the *NPH3::GFP-NPH3* construct, transiently expressed in the leaves of *Nicotiana benthamiana* and compared with the expression of the non-mutated GFP-NPH3 control. Transfected *N. benthamiana* plants were dark-adapted before confocal observation. The localisation of transiently expressed GFP-NPH3 was similar to that of functionally active GFP-NPH3 in *Arabidopsis*[28] described above, and repeated scanning with the 488-nm laser used to concomitantly excite GFP along with endogenous phot1, induced relocalisation of GFP-NPH3 into aggregates (Fig. 2b). The localisation of each of the transiently expressed NPH3 phospho-mutants was the same as GFP-NPH3 when imaged immediately (scan 1). Repeated laser scanning was effective in inducing relocalisation for both

NPH3-M1 and M2 constructs, whereas the NPH3-M3 mutant failed to show any light-induced changes in subcellular localisation.

Phot1-induced changes in NPH3 localisation are correlated with changes in NPH3 phosphorylation status in transgenic *Arabidopsis* seedlings[27,28]. Immunoblot analysis of protein extracts from dark-adapted leaves of *N. benthamiana* transiently expressing GFP-NPH3 irradiated with blue light also showed enhanced electrophoretic mobility compared with leaves maintained in darkness (Fig. 2c), although to a lesser degree than observed in etiolated *Arabidopsis* seedlings expressing GFP-NPH3 when equivalent light treatments were used (Fig. 1b). Both the NPH3-M1 and M3 mutants were affected for this response, whereas the NPH3-M2 mutant response was similar to GFP-NPH3 (Fig. 2c). The NPH3-M1 mutant showed enhanced electrophoretic mobility in the dark compared with the GFP-NPH3 construct, with a further slight enhancement following blue light treatment. The NPH3-M1 mutant contains mutations of serine residues S213, S223, S233 and S237, mutation of which was previously shown to contribute to reducing the electrophoretic mobility of NPH3 in darkness[29]. Conversely, the NPH3-M3 mutant migrated at the same position as GFP-NPH3 in the dark, even following blue light irradiation. Therefore, mutation of phosphorylation sites within the M3 region at the C-terminus of NPH3 prevents both blue light-induced dephosphorylation of sites within the NPH3 domain, which contribute to reducing the electrophoretic mobility, as well as subcellular relocalisation into aggregates.

The C-terminal amino-acid sequence of NPH3 is highly conserved in angiosperms (Supplementary Fig. 1a) and contains two serine residues, S744 and S746 in *Arabidopsis* NPH3. Mutation of either serine residue to alanine, singularly or together, prevented (for S744A and S744A S746A) or greatly reduced (for S746A) the light-induced relocalisation response when transiently expressed in *N. benthamiana* (Fig. 2d). Similarly, these mutations also prevented dephosphorylation of NPH3 following blue light irradiation (Fig. 2e). Therefore, mutation of S744 and/or S746 can reproduce the results obtained with the NPH3-M3 mutant. Although serine to alanine mutations effectively blocks phosphorylation of the respective residue, phosphomimetic substitutions aim to mimic the phosphorylated state by replacement with a negatively charged amino acid. However, mutation of S744 and S746 to aspartate produced similar results to the alanine mutations; loss of light-induced relocalisation and dephosphorylation (Supplementary Fig. 1b, Supplementary Fig. 1c).

**S744 is required for 14-3-3 binding and early signalling events.** To examine the effects of the C-terminal serine residues S744 and S746 in NPH3 signalling, we generated transgenic *Arabidopsis* expressing *NPH3::GFP-NPH3* containing S744A S746A, S744D S746D, S744A or S746A mutations in the *nph3* mutant background. Confocal imaging of hypocotyl cells of etiolated seedlings expressing NPH3 S744A S746A or NPH3 S744D S746D showed that both mutants did not relocalise into aggregates following irradiation with the 488 nm laser, in contrast to the GFP-NPH3 control (Fig. 3a). The single NPH3 S744A mutant also lacked this response, whereas the NPH3 S746A mutant was unaffected. Furthermore, analysis of NPH3 dephosphorylation showed that seedlings expressing NPH3 S744A S746A, S744D S746D or S744A exhibited no change in electrophoretic mobility with blue light treatment, in contrast to NPH3 S746A mutant and GFP-NPH3 expressing lines, which both displayed enhanced mobility with blue light treatment (Fig. 3b). Whereas results from transient expression analysis in *N. benthamiana* showed both S744 and S746 were involved in these early signalling responses (Fig. 2d, e), analysis of transgenic *Arabidopsis* identifies only S744 as being required.

To determine whether S744 was also required to mediate interactions between NPH3 and 14-3-3 proteins, far-western blotting was performed on anti-GFP IPs from *nph3* seedlings

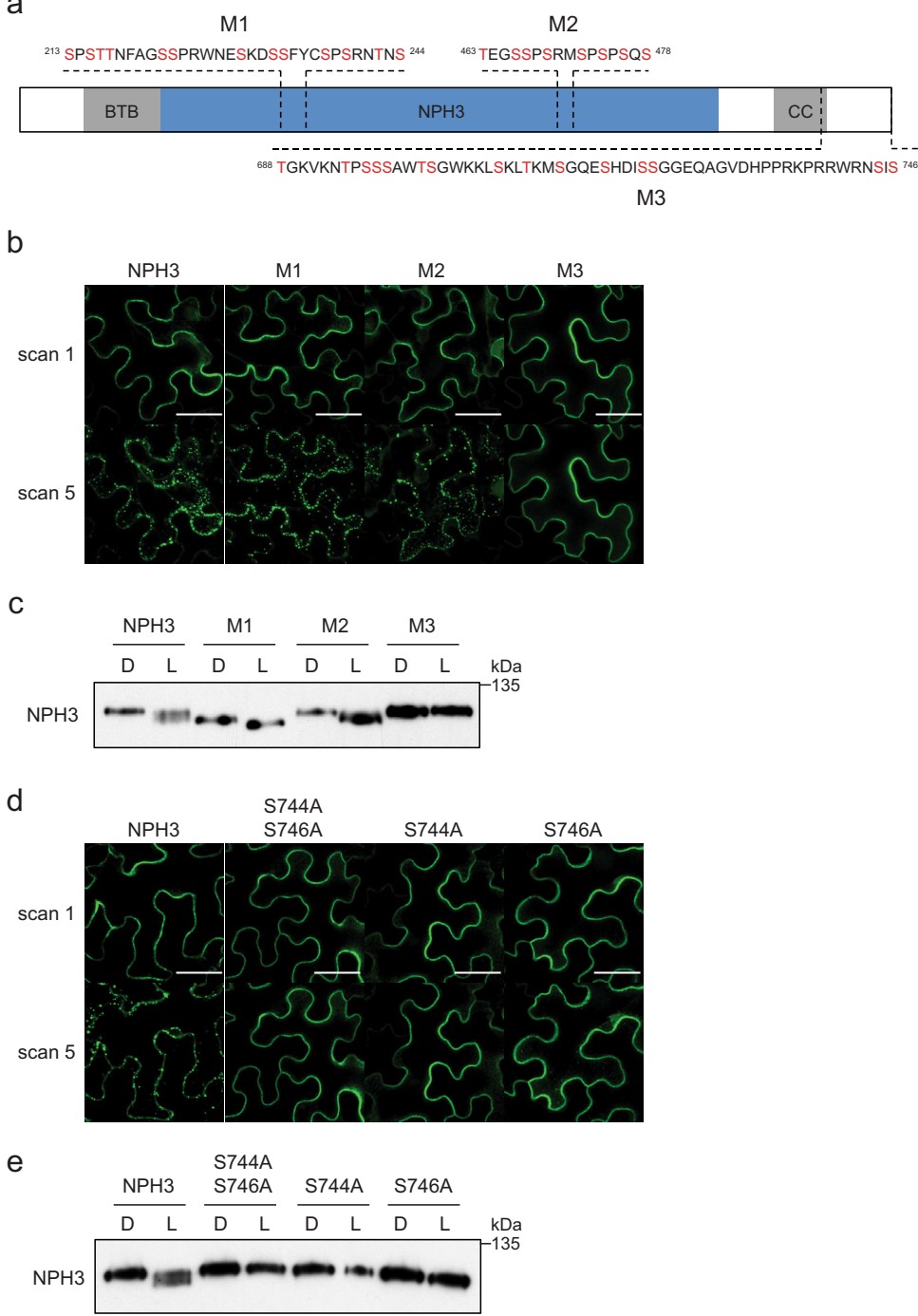

**Fig. 2 Mutational analysis of NPH3 phosphorylation sites. a** Schematic illustration of the NPH3 protein indicating the location of the three mutagenized regions (M1, M2 and M3). For each region, all serine and threonine residues in red were substituted with alanine. The relative positions of the bric-a-brac, tramtrack and broad complex (BTB), NPH3 and coiled-coil (CC) domains are indicated. **b** Confocal images of GFP-NPH3 (NPH3) and phosphorylation site mutants (M1, M2 and M3) transiently expressed in leaves of *N. benthamiana*. Plants were dark-adapted before confocal observation and images acquired immediately (scan 1) and after repeat scanning with the 488 nm laser (scan 5). Bar, 50 μm. **c** Immunoblot analysis of protein extracts from leaves of *N. benthamiana* transiently expressing GFP-NPH3 (NPH3) and phosphorylation site mutants (M1, M2 and M3). Plants were dark-adapted and maintained in darkness (D) or irradiated with 20 μmol m$^{-2}$ s$^{-1}$ of blue light for 15 min (L). Protein extracts were probed with anti-GFP antibodies. **d** Confocal images of GFP-NPH3 (NPH3) and phosphorylation site mutants S744A S746A, S744A and S746A transiently expressed in leaves of *N. benthamiana*. Plants were dark-adapted before confocal observation and images acquired immediately (scan 1) and after repeat scanning with the 488 nm laser (scan 5). Bar, 50 μm. **e** Immunoblot analysis of protein extracts from leaves of *N. benthamiana* transiently expressing GFP-NPH3 (NPH3) and phosphorylation site mutants S744A S746A, S744A and S746A. Plants were dark-adapted and maintained in darkness (D) or irradiated with 20 μmol m$^{-2}$ s$^{-1}$ of blue light for 15 min (L). Protein extracts were probed with anti-GFP antibodies. Experiments were repeated at least twice with similar results.

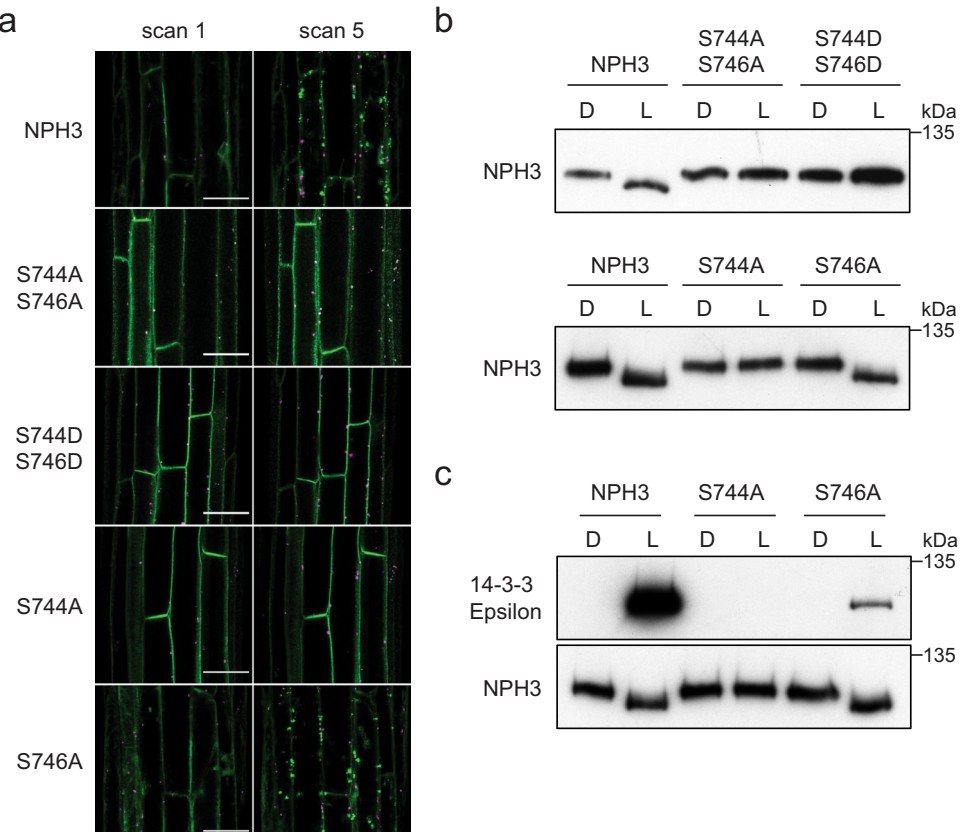

**Fig. 3 S744 is required for 14-3-3 binding and early signalling events. a** Confocal images of hypocotyl cells of etiolated *nph3* seedlings expressing GFP-NPH3 (NPH3) or phosphorylation site mutants S744A S746A, S744D S746D, S744A and S746A. Seedlings were scanned immediately (scan 1) and again after repeat scanning with the 488 nm laser (scan 5). GFP is shown in green and autofluorescence in magenta. Bar, 50 μm. **b** Immunoblot analysis of total protein extracts from etiolated *nph3* seedlings expressing GFP-NPH3 (NPH3) or phosphorylation site mutants maintained in darkness (D) or irradiated with 20 μmol m$^{-2}$ s$^{-1}$ blue light for 15 min (L). Protein extracts were probed with anti-NPH3 antibodies. **c** Far-western blot analysis of anti-GFP immunoprecipitations from etiolated *nph3* seedlings expressing GFP-NPH3 (NPH3) or phosphorylation site mutants (S744A or S746A) maintained in darkness (D) or irradiated with 20 μmol m$^{-2}$ s$^{-1}$ of blue light for 15 min blue light (L). GST-tagged 14-3-3 isoform Epsilon was used as the probe. Blots were probed with an anti-NPH3 antibody as a loading control (bottom panel). Experiments were repeated at least twice with similar results.

expressing GFP-NPH3 or GFP-NPH3 containing S744A or S746A mutations (Fig. 3c). The binding of recombinant 14-3-3 Epsilon was evident for both GFP-NPH3 and NPH3 S746A mutant in a light-dependent manner, with the signal for S746A being substantially lower. However, no binding could be detected for the NPH3 S744A mutant. Phosphorylation of S744 is, therefore, necessary for 14-3-3 binding, subcellular relocalisation and dephosphorylation of N-terminal sites (including S213, S223, S233 and S237) in response to blue light perception.

**Phot1 phosphorylates NPH3 at position S744 in a light-dependent manner.** Given the evidence for light-induced phosphorylation of NPH3, we examined whether NPH3 was a direct substrate for phot1-kinase activity using a gatekeeper engineered phot1 (phot1$^{GK}$), which can accommodate the bulky ATP analogue N$^6$-benzyl-ATPγS as a thiophospho-donor[20]. NPH3, or the NPH3 S744A mutant, were co-expressed in a cell-free expression system with phot1$^{GK}$ and used for in vitro kinase assays in the presence of N$^6$-benzyl-ATPγS. Light-induced thiophosphorylation, which can be detected by immunoblotting with anti-thiophoshoester antibody following chemical alkylation of the incorporated thiophosphates, was detected for NPH3 but not for the NPH3 S744A mutant (Fig. 4a), showing phot1 can specifically phosphorylate residue S744 of NPH3 in vitro. To detect the phosphorylation status of S744 in vivo, we raised a phospho-specific antibody (pS744).

Phosphorylation of S744 was observed in WT and GFP-NPH3-expressing seedlings in a light-dependent manner and mutation of S774 resulted in a loss of signal, demonstrating the specificity of the pS744 phospho-specific antibody (Fig. 4b). Phosphorylation of S744 was also detectable for NPH3 S746A mutant-expressing seedlings at a reduced level, similar to the results observed for 14-3-3 binding (Fig. 3c).

Phot1 is the main photoreceptor mediating phototropism to low (<1 μmol m$^{-2}$ s$^{-1}$) and high (>1 μmol m$^{-2}$ s$^{-1}$) fluence rates of blue light, whereas phot2 functions predominantly at higher light intensities[3] (>10 μmol m$^{-2}$ s$^{-1}$). Phosphorylation of S744 occurred in WT seedlings in response to both low blue (0.5 μmol m$^{-2}$ s$^{-1}$) and high blue (50 μmol m$^{-2}$ s$^{-1}$) light treatments concomitantly with dephosphorylation of sites within the NPH3 domain, detected via changes in electrophoretic mobility when probed with anti-NPH3 antibody (Fig. 4c). These responses were absent in *phot1 phot2* double mutant and *phot1* single mutant seedlings, but unchanged in the *phot2* single mutant, demonstrating that phosphorylation of S744 and dephosphorylation of sites within the NPH3 domain, which alter electrophoretic mobility, are phot1-specific responses in etiolated seedlings.

To assess the kinetics of changes in NPH3 phosphorylation status, we performed time-course experiments. Dephosphorylation of sites that alter NPH3 electrophoretic mobility required 15 min of blue light irradiation (Fig. 5a), whereas phosphorylation of S744 was detected within 30 s and maintained over the 2 h

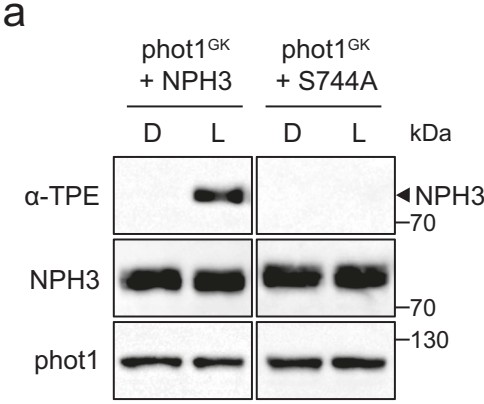

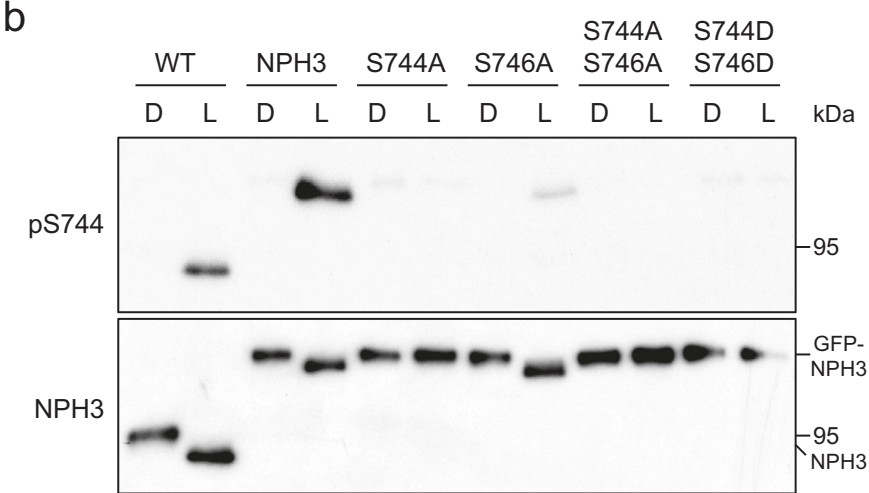

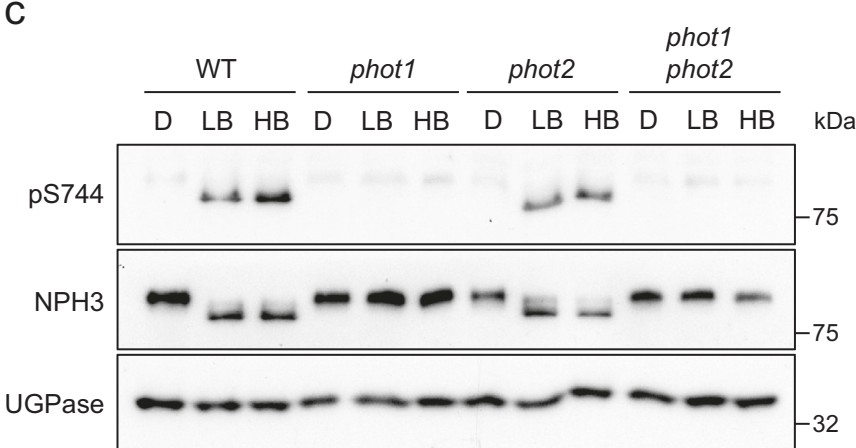

**Fig. 4 Phot1 phosphorylates NPH3 at position S744 in a light-dependent manner. a** Thiophosphorylation analysis of in vitro kinase assays containing gatekeeper engineered phot1 (phot1$^{GK}$) and NPH3 or NPH3 S744A. Reactions were performed in the absence (D) or presence of 20 s of white light (L), and thiophosphorylation was detected using anti-thiophoshoester antibody (α-TPE). Blots were probed with anti-NPH3 and anti-phot1 antibodies. **b** Immunoblot analysis of total protein extracts from etiolated wild-type (WT) or *nph3* seedlings expressing GFP-NPH3 (NPH3) or phosphorylation site mutants maintained in darkness (D) or irradiated with 20 µmol m$^{-2}$ s$^{-1}$ blue light for 15 min (L). Protein extracts were probed with phospho-specific pS744 antibody and anti-NPH3 antibodies. **c** Immunoblot analysis of total protein extracts from etiolated wild-type (WT) or *phot1*, *phot2* and *phot1 phot2* mutant seedlings maintained in darkness (D) or irradiated with 0.5 µmol m$^{-2}$ s$^{-1}$ (low blue; LB) or 50 µmol m$^{-2}$ s$^{-1}$ (high blue; HB) of blue light for 60 min. Blots were probed with phospho-specific pS744 antibody, anti-NPH3 anti-UGPase (loading control) antibodies. Experiments were repeated at least twice with similar results.

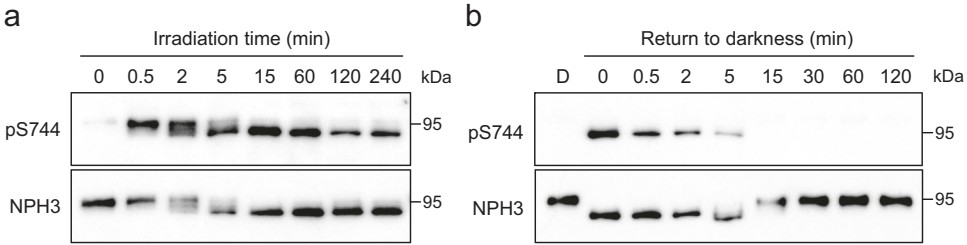

**Fig. 5 Kinetics of phot1-mediated phosphorylation of NPH3. a** Time-course of S744 phosphorylation. Immunoblot analysis of total protein extracts from etiolated wild-type seedlings irradiated with 0.5 μmol m$^{-2}$ s$^{-1}$ of blue light for the time indicated. Blots were probed with anti-pS744 and anti-NPH3 antibodies. **b** Time-course of S744 dephosphorylation. Immunoblot analysis of total protein extracts from etiolated wild-type seedlings maintained in darkness (D) or irradiated with 0.5 μmol m$^{-2}$ s$^{-1}$ for 15 min and returned to darkness for the time indicated. Blots were probed with anti-pS744 and anti-NPH3 antibodies. Experiments were repeated at least twice with similar results.

irradiation period. When etiolated seedlings were returned to darkness following blue light exposure, S744 was dephosphorylated within 15 min, matching the time required for rephosphorylation of sites responsible for the electrophoretic mobility shift (Fig. 5b). Therefore, phot1 phosphorylation of S744 is rapid, occurring before light-induced dephosphorylation of sites within the NPH3 domain, and reversible in darkness.

**Phot1 phosphorylation of NPH3 promotes functionality.** *Arabidopsis* mutants lacking NPH3 fail to exhibit hypocotyl phototropism under a variety of different light conditions[33,46]. Phototropism in two independent homozygous transgenic *nph3* mutants expressing *NPH3::GFP-NPH3* is restored to levels comparable to non-transgenic WT seedlings when irradiated with 0.5 μmol m$^{-2}$ s$^{-1}$ of unilateral blue light (Fig. 6a). In contrast, the magnitude and kinetics of phototropic curvature were reduced in seedlings expressing GFP-NPH3 with both S744 and S746 residues mutated to alanine or aspartate (Fig. 6a, Supplementary Fig. 2a). Similarly, phototropism was reduced in seedlings expressing GFP-NPH3 containing the S744A mutation, while the NPH3 S746A mutant-expressing seedlings were fully functional (Fig. 6b, Supplementary Fig. 2b). To determine whether the reduced phototropic responsiveness of the S744A mutant is due to altered photosensitivity, phototropism was further assessed under lower (0.05 μmol m$^{-2}$ s$^{-1}$; Fig. 6c, Supplementary Fig. 2c) and higher (20 μmol m$^{-2}$ s$^{-1}$; Fig. 6d, Supplementary Fig. 2d) intensity blue light irradiation. Under both fluence rates, transgenic lines expressing the NPH3 S744A mutant were less responsive than the GFP-NPH3 or NPH3 S746A mutant-expressing lines.

NPH3 also functions in phototropin-mediated leaf positioning, particularly in low light environments[25]. In WT seedlings transferred to low-intensity white light (10 μmol m$^{-2}$ s$^{-1}$) the petioles of the first true leaves were positioned obliquely upwards in order to maximise light capture, while the petioles of *nph3* mutant seedlings were positioned horizontally (Fig. 6e). Seedlings expressing GFP-NPH3 or the NPH3 S746A mutant were complemented for petiole positioning, while the response of seedlings expressing the NPH3 S744A mutant was significantly reduced (Fig. 6e), which was also observed for the NPH3 S744A S746A and S744D S746D transgenic lines (Supplementary Fig. 3). These results demonstrate that phot1 phosphorylation of S744 positively regulates NPH3 function.

**Phosphorylation and 14-3-3 binding drives NPH3 relocalisation.** The phenotypes of seedlings expressing GFP-NPH3 containing S744D S746D mutations were identical to seedlings expressing NPH3 with the S744A S746A mutations (Fig. 3, Fig. 6a), consistent with reports that aspartate does not effectively mimic phosphorylation with respect to 14-3-3 binding[37,47]. To create a constitutively 14-3-3 binding variant, the sequence

encoding the last three amino acids of the *NPH3::GFP-NPH3* construct, including the S744 phosphorylation site, was replaced with the R18 peptide sequence (Fig. 7a). R18 is a synthetic peptide that mediates phosphorylation-independent binding of 14-3-3 proteins with high affinity[48]. As a control, a construct containing a mutated version of the R18 sequence (mR18), known to abolish 14-3-3 binding[49], was also generated (Fig. 7a). When transiently expressed in *N. benthamiana*, GFP-NPH3-R18 appeared as aggregates when dark-adapted leaves were imaged immediately, with no change in localisation during imaging (Fig. 7b). Conversely, GFP-NPH3-mR18 remained localised to the plasma membrane following repeated laser scanning, as previously observed with GFP-NPH3 constructs lacking the S744 phosphorylation site (Fig. 2d, Supplementary Fig. 1b).

To confirm these results in stable transgenic lines, *Arabidopsis nph3* mutants were transformed with *NPH3::GFP-NPH3* containing the R18 or mR18 sequences. Confocal imaging of hypocotyl cells of etiolated seedlings revealed similar patterns of localisation observed in *N. benthamiana*, with GFP-NPH3-R18 forming aggregates in darkness, whereas GFP-NPH3-mR18 failed to relocalise following repeated laser scanning (Fig. 7c). Consistent with the subcellular localisation patterns, analysis of NPH3 dephosphorylation showed that lines expressing GFP-NPH3-mR18 display no change in electrophoretic mobility following blue light treatment, whereas a portion of GFP-NPH3-R18 exhibited enhanced electrophoretic mobility both in darkness and after irradiation (Fig. 7d). Far-western blotting was used to confirm the constitutive binding of recombinant 14-3-3 Epsilon to GFP-NPH3-R18 immunoprecipitated from seedlings maintained in darkness and following blue light irradiation, as well as the absence of 14-3-3 binding to GFP-NPH3-mR18 (Fig. 7e). Together, these results show that engineered 14-3-3 binding, independent from phot1-mediated S744 phosphorylation, is partially sufficient to induce changes in NPH3 dephosphorylation and localisation status.

To assess functionality, phototropism was measured in GFP-NPH3-R18 and GFP-NPH3-mR18-expressing seedlings irradiated with 0.05 μmol m$^{-2}$ s$^{-1}$ (Fig. 7f, Supplementary Fig. 4a), 0.5 μmol m$^{-2}$ s$^{-1}$ (Fig. 7g, Supplementary Fig. 4b) or 20 μmol m$^{-2}$ s$^{-1}$ (Fig. 7h, Supplementary Fig. 4c) of unilateral blue light. Phototropic responsiveness was reduced under all fluence rates for GFP-NPH3-mR18-expressing seedlings (Fig. 7f–h, Supplementary Fig. 4a–c), matching the phenotype of seedlings expressing GFP-NPH3 containing the S744A mutation (Fig. 6b–d). Phototropism was further reduced in seedlings expressing GFP-NPH3-R18 (Fig. 7f–h, Supplementary Fig. 4a–c), which also displayed an increased variability in the direction of curvature (Supplementary Fig. 5) compared with the GFP-NPH3-mR18 lines. Therefore, while NPH3 mutants unable to bind 14-3-3 proteins have a reduced ability to re-orientate growth towards a light source, constitutively

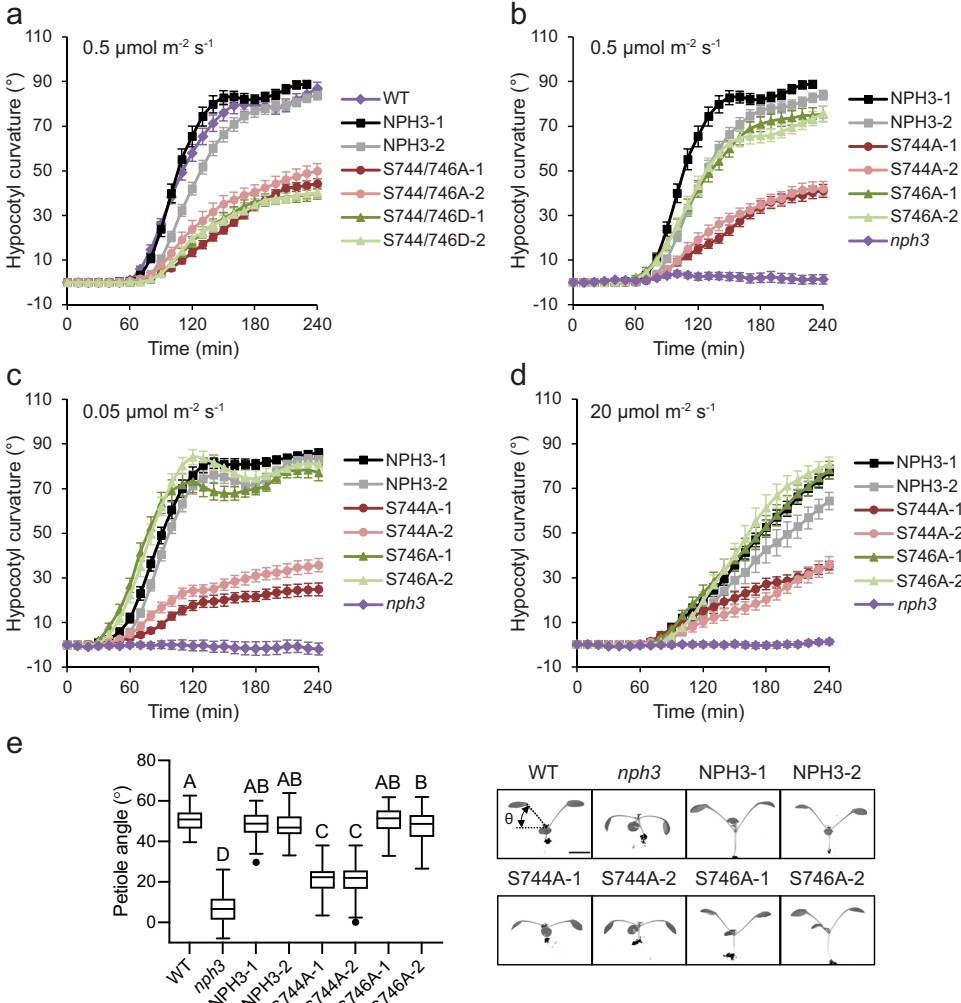

**Fig. 6 Phot1 phosphorylation of NPH3 promotes functionality. a** Phototropism of etiolated wild-type (WT) seedlings, *nph3* seedlings expressing GFP-NPH3 (NPH3) or phosphorylation site mutants S744A S746A and S744D S746D irradiated with 0.5 μmol m$^{-2}$ s$^{-1}$ unilateral blue light. **b–d** Phototropism of etiolated *nph3* seedlings expressing GFP-NPH3 (NPH3), phosphorylation site mutants S744A or S746A and *nph3* mutant seedling irradiated with **b** 0.5 μmol m$^{-2}$ s$^{-1}$, **c** 0.05 μmol m$^{-2}$ s$^{-1}$ or **d** 20 μmol m$^{-2}$ s$^{-1}$ unilateral blue light. Hypocotyl curvatures were measured every 10 min for 4 h, and each value is the mean ± SE of 17–20 seedlings from two independent biological replicates. Exact *n* values are provided in the Source data file. **e** Petiole positioning of WT, *nph3* mutant and *nph3* seedlings expressing GFP-NPH3 or phosphorylation site mutants S744A and S746A. Plants were grown under 80 μmol m$^{-2}$ s$^{-1}$ white light for 9 d before transfer to 10 μmol m$^{-2}$ s$^{-1}$ white light for 5 d. Box and whisker plots show petiole angle from the horizontal measured for the first true leaves. The centre line indicates the median, the bounds of the boxes indicate the 25th and 75th percentiles, whiskers extend 1.5 times the interquartile range and outliers are represented by dots. (*n* = 60 seedlings, from three independent biological replicates). Different letters denote significant differences (*P* < 0.01, one-way ANOVA with Tukey post test). Exact *P* values are provided in the Source data file. Representative images for each genotype are shown on the right. Bar, 5 mm.

14-3-3 bound NPH3 mutants also display a diminished ability to sense the directionality of a light source.

## Discussion

In this study, we used MS to identify proteins co-immunoprecipitating with GFP-NPH3. This revealed 14-3-3 proteins as NPH3 interactors specifically following a blue light treatment (Fig. 1). Using a chemical-genetic approach, we have found that NPH3 is phosphorylated by phot1 on the C-terminally positioned S744 in a light-dependent manner (Fig. 4a). Moreover, the generation of anti-pS744 antibodies confirmed light-induced phosphorylation of S744 in vivo (Fig. 4c). Phototropins are members of the AGCVIII (protein kinase A, cyclic GMP-dependent protein kinase and protein kinase C) subfamily of protein kinases[50] and S744 is part of a PKA-like phosphorylation consensus sequence (RxS), as are the previously

identified phot1-kinase substrates BLUS1[15], CBC1[16] and PKS4[19] (Supplementary Fig. 6a).

Phot1-mediated phosphorylation of S744 is required to elicit the previously documented early cellular events associated with NPH3 activation such as dephosphorylation[30] and subcellular relocalisation[27,28]. This is consistent with previous observations of changes in NPH3 electrophoretic mobility correlating with the life-time duration of phot1 activation in planta[7] and occurring locally only in cells/tissues where both proteins are present[51]. Furthermore, a constitutively active phot1-variant can induce NPH3 dephosphorylation in darkness[52]. The phosphorylation status of residues S213, S223, S233 and S237 contribute to reducing the electrophoretic mobility of NPH3 in darkness[29], however other unidentified sites are also involved (Fig. 2c;[27]). Recently, phosphopeptide mapping of YFP tagged NPH3 immunoprecipitated from etiolated seedlings maintained in darkness or irradiated with blue light identified seven

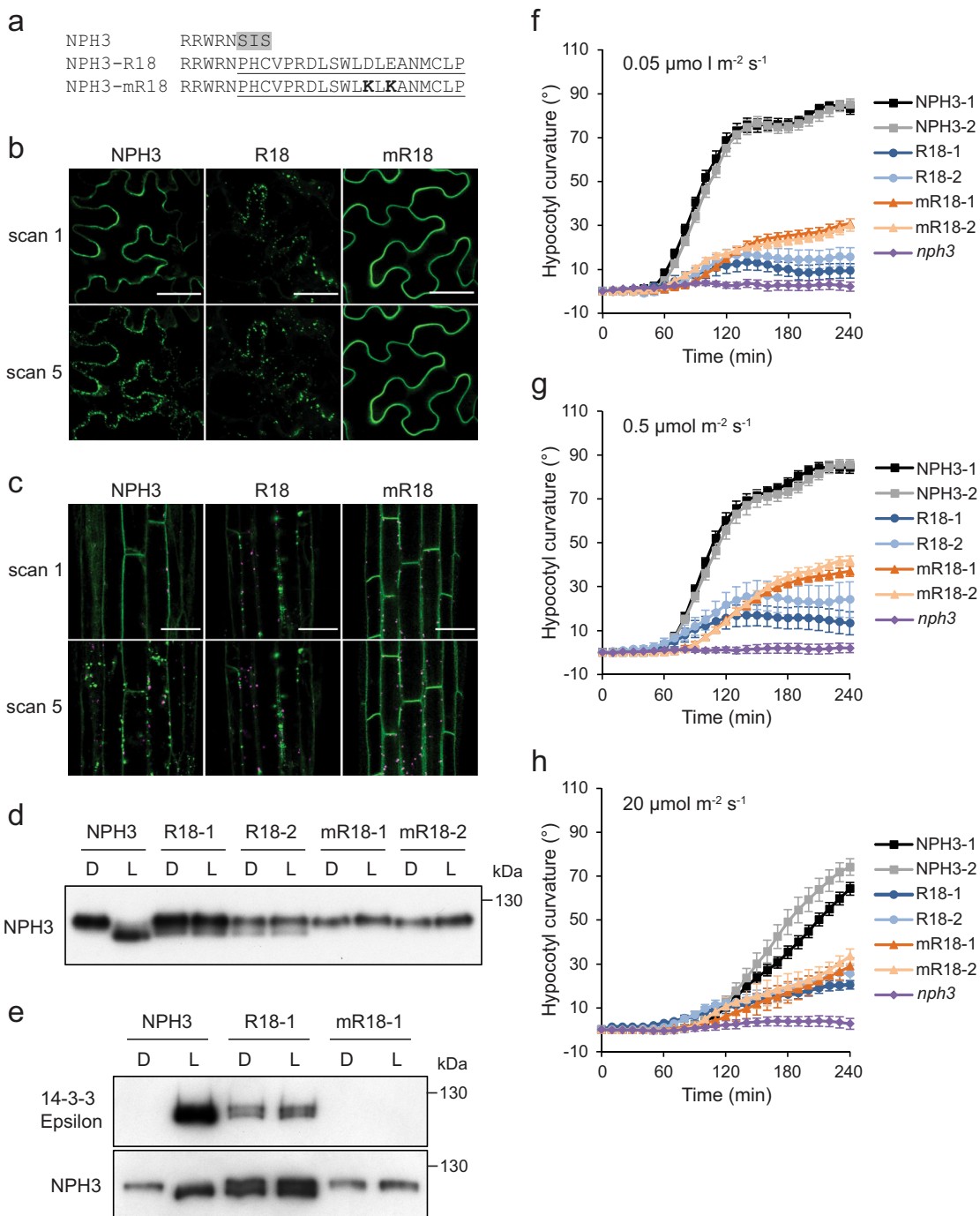

**Fig. 7 Analysis of a constitutive 14-3-3 binding NPH3 variant. a** Amino-acid sequence of the NPH3-R18 and mR18 constructs. Residues 744–746 of NPH3 (grey shaded) were replaced with the R18 peptide sequence (underlined). Two lysine residues (bold) were introduced into the mR18 sequence to abolish 14-3-3 binding. **b**, **c** Confocal images of GFP-NPH3 (NPH3), GFP-NPH3 containing the R18 peptide sequence (R18) or the mutated R18 peptide sequence (mR18) (**b**) transiently expressed in leaves of *N. benthamiana* plants, dark-adapted before confocal observation and (**c**) in hypocotyl cells of etiolated transgenic *Arabidopsis nph3* seedlings. Images were acquired immediately (scan 1) and after repeat scanning with the 488 nm laser (scan 5). GFP is shown in green and autofluorescence in magenta. Bar, 50 μm. These experiments were repeated at least twice with similar results. **d** Immunoblot analysis of total protein extracts from etiolated *nph3* seedlings expressing NPH3, R18 or mR18 maintained in darkness (D) or irradiated with 20 μmol m$^{-2}$ s$^{-1}$ blue light for 15 min (L). Protein extracts were probed with anti-NPH3 antibodies. This experiment was repeated twice with similar results. **e** Far-western blot analysis of anti-GFP immunoprecipitations from etiolated *nph3* seedlings expressing NPH3, R18 or mR18 maintained in darkness (D) or irradiated with 20 μmol m$^{-2}$ s$^{-1}$ of blue light for 15 min blue light (L). GST-tagged 14-3-3 isoform Epsilon was used as the probe. Blots were probed with anti-NPH3 antibody as a loading control (bottom panel). Phototropism of etiolated *nph3* seedlings expressing GFP-NPH3 (NPH3), R18 or mR18 and *nph3* mutant seedling irradiated with **f** 0.05 μmol m$^{-2}$ s$^{-1}$, **g** 0.5 μmol m$^{-2}$ s$^{-1}$ or **h** 20 μmol m$^{-2}$ s$^{-1}$ of unilateral blue light. Hypocotyl curvatures were measured every 10 min for 4 h, and each value is the mean ± SE of 18–20 seedlings from two independent biological replicates. Exact *n* values are provided in the Source data file.

phosphorylation sites, including S213, S223 and S237[53]. However, differential phosphorylation was evident for only two phosphorylation sites, phosphorylated S213 and S237 was detected in samples from seedlings maintained in darkness and absent following blue light irradiation[53]. The kinase(s) and phosphatase(s) regulating the phosphorylation status of these sites are currently unknown and their role in regulating NPH3 signalling is unclear. However, mutation of S213, S223, S233 and S237 to alanine, or deletion of amino-acid residues S213–S239, did not impact their ability to restore phototropism in *nph3* mutant seedlings[29], or form aggregates when transiently expressed in *N. benthamiana* (Fig. 2b).

Phosphorylation of S744 creates a 14-3-3 binding site (Fig. 3c), which conforms to the C-terminal mode III 14-3-3-binding motif pS/pTX$_{1-2}$-COOH[38]. We created a translational fusion between NPH3 and the synthetic R18 peptide to study the role of 14-3-3 binding in the absence of phot1 phosphorylation (Fig. 7e). 14-3-3 binding alone was able to induce NPH3 relocalisation into aggregates (Fig. 7b, c) and partially reduce the electrophoretic mobility of NPH3 (Fig. 7d), in the absence or presence of light. Light-dependent 14-3-3 binding has also been shown for phot1; non-epsilon 14-3-3 s bind to 3 phosphorylation sites located between the LOV1 and LOV2 photosensory domains[32], but the functional relevance of this interaction is unknown as mutation of 2 of the phosphorylation sites did not impair functionality[25]. In contrast, no isoform specificity was observed for 14-3-3 binding to NPH3, with both epsilon and non-epsilon isoforms shown to interact (Fig. 1). Functional redundancy between 14-3-3 isoforms means loss-of-function mutants often show few, if any, phenotypes, with even quadruple non-epsilon 14-3-3 mutants displaying mild growth phenotypes under non-stress growth conditions[54], with no obvious differences in phototropism or NPH3 dephosphorylation kinetics observed compared with WT seedlings (Supplementary Fig. 7). However, conditional RNA interference lines targeting three 14-3-3 epsilon members (epsilon, mu and omicron) displayed several auxin-related phenotypes, including reduced hypocotyl elongation and defects in root and hypocotyl gravitropism, due to altered polarity of the PIN-FORMED (PIN) auxin transporters as a consequence of 14-3-3 regulation of cellular trafficking[42]. NPH3 is also reported to be required for phot1-driven changes in PIN2 trafficking during negative phototropic bending of roots[55].

The biochemical basis underpinning phototropism is the formation of a gradient of phot1 activation across the stem[56], which results in an asymmetric accumulation of auxin on the shaded side through an unidentified mechanism[2]. We previously demonstrated that a gradient of GFP-NPH3 relocalisation occurs across the hypocotyl of *Arabidopsis* seedlings during unilateral irradiation with blue light[28]. Here, we report that seedlings expressing mutants of GFP-NPH3 unable to form such a gradient, either through mutation of the phosphorylation site required for 14-3-3 binding (S744) or owing to constitutive 14-3-3 binding via the R18 peptide, have a severely compromised phototropic response. Thus, phototropic curvature likely involves signalling outputs mediated by a gradient in NPH3 localisation across the stem. Our current findings are therefore consistent with 14-3-3 proteins being instrumental components regulating auxin-dependent growth[42].

The phot1 phosphorylation consensus sequence of NPH3 is also conserved in several other NRL proteins including RPT2, NCH1, and members of the NAKED PINS IN YUCCA (NPY) clade (Supplementary Fig. 6b). Notably, RPT2 was identified in immunoprecipitants of seedlings expressing 14-3-3 epsilon-GFP[42]. We could also detect phosphorylation of RPT2 on the corresponding serine residue (S591) when co-expressed with phot1$^{GK}$ in vitro kinase assays (Supplementary Fig. 6c), as well as light-dependent phosphorylation of DEFECTIVELY ORGANISED TRIBUTARIES 3 (DOT3), NPY1 and NRL1 (Supplementary Fig. 6d). It is therefore possible the residual functionality seen in GFP-NPH3 S744A seedlings (Fig. 6) arises

from co-action with other NRL family members. The NPY clade of NRL proteins function redundantly to mediate organogenesis and root gravitropism[57–59]. These responses are not reported to involve phototropin signalling, but rather the related AGCVIII kinases PINOID (PID) and its close homologues WAG1 and WAG2, and the D6 PROTEIN KINASE (D6PK) family[60]. PID/WAGs and D6PKs phosphorylate PIN transporters on RxS phosphorylation site motifs[50] and physically interact with NPY proteins to maintain PIN polar localisation and therefore directional auxin transport[60]. Furthermore, aggregate formation is not limited to NPH3 but has also been documented for NPY1 when expressed in *Arabidopsis* protoplasts[57]. Therefore, phosphorylation and concomitant 14-3-3 binding to the C-terminus may represent a conserved mechanism of regulation for NRL proteins.

Determining the biochemical function of NPH3, and related family members, is now required to understand how phots signal via NRL proteins to coordinate different light-capturing processes in plants that will ultimately offer new opportunities to manipulate plant growth through alterations in photosynthetic capacity.

## Methods

**Plant material and growth.** Wild-type *Arabidopsis* (*gl-1*, ecotype Columbia), *nph3-6*[21], 14-3-3 quadruple mutants[54] and the GFP-NPH3 transgenic line[28] were previously described. Unless otherwise stated, seeds were sown on soil or surface sterilised and plated on half-strength Murashige and Skoog (MS) medium with 0.8% agar (w/v) and stratified at 4 °C for 2–5 d. Seeds on soil were transferred to a controlled environment room (Fitotron, Weiss Technik) with LED illumination (C65NS12, Valoya) under 16 h 22 °C/8 h 18 °C light: dark cycles and 80 μmol m$^{-2}$ s$^{-1}$ white light. Seeds on MS medium were exposed to 80 μmol m$^{-2}$ s$^{-1}$ white light for 6–8 h to induce germination and grown vertically in darkness for 3 d. For blue light treatment, white light was filtered through Moonlight Blue filter No. 183 (Lee Filters). Fluence rates for all light sources were measured with a Li-250A and quantum sensor (LI-COR).

**Transient expression in *Nicotiana benthamiana*.** To create transformation vectors encoding *NPH3* with multiple serine and threonine residues mutated to alanine, fragments of *NPH3* were synthesised (ThermoFisher Scientific) encoding the 13 alanine substitutions for *NPH3-M1*, 8 substitutions for *NPH3-M2* and 15 substitutions for *NPH3-M3*. The synthesised fragments were introduced into *NPH3::GFP-NPH3* using *KpnI* and *MluI* restriction sites for *GFP-NPH3-M1*, *MluI* and *PstI* restriction sites for *GFP-NPH3-M2* and *PstI* and *BamHI* restriction sites for *GFP-NPH3-M3*. *Agrobacterium*-mediated transient expression in *Nicotiana benthamiana* was performed as reported previously[61]. *Agrobacterium tumefaciens* strain GV3101, transformed with the plasmid of interest, was resuspended in infiltration buffer (10 mM MgCl$_2$, 10 mM MES-KOH [pH 5.6] and 200 mM acetosyringone) at an OD$_{600}$ of 0.4 and syringe-infiltrated into leaves of 3–4-week-old *N. benthamiana* plants. Plants were dark-adapted for 16 h before 1 cm leaf discs for confocal observation or protein extraction were taken 2 d post infiltration. For blue light irradiation, leaf discs were placed abaxial-side upwards on the surface of MS medium agar plates for the duration of the treatment.

**Transformation of *Arabidopsis*.** Amino-acid substitutions of S744 and/or S746 were introduced into the pUC-SP vector containing the *NPH3*-coding sequence by site-directed mutagenesis and verified by DNA sequencing. The coding sequence of *NPH3* in the *NPH3::GFP-NPH3* pEZR(K)-LC binary vector[28] was replaced by the coding sequences containing the phosphosite mutations using Gibson Assembly (New England Biolabs). To create transformation vectors *NPH3::GFP-NPH3-R18* and *NPH3::GFP-NPH3-mR18*, a fragment encoding amino-acid residues 419–743 was PCR amplified from *NPH3* pUC-SP with primers containing the R18 or mR18-coding sequence and inserted into *NPH3::GFP-NPH3* using *MluI* and *BamHI* restriction sites. All primer sequences are available in Supplementary Table 1. The *nph3-6* mutant was transformed with *Agrobacterium tumefaciens* strain GV3101 using a streamlined floral dipping protocol[62]. A 500 ml saturating *A. tumefaciens* culture, transformed with the plasmid of interest, was grown in YEBS medium at 28 °C in a shaking incubator, diluted with 500 ml of 5% (w/v) sucrose and Silwet® L-77 added to a final concentration of 0.01% (v/v). Flowering *nph3-6* mutant plants were briefly dipped into the solution and sealed in a plastic bag overnight. Plants were dipped for a second time 3–5 days later. Based on the segregation of kanamycin resistance independent homozygous T3 lines, or for GFP-NPH3-mR18 transgenics single-insertion T2 lines, were selected for analysis.

**Phototropism.** Phototropism was performed using free-standing etiolated seedlings[51]. Seeds were sown in transparent plastic entomology boxes (Watkins and Doncaster) on a layer of silicon dioxide (Honeywell, Fluka), watered with quarter-strength MS medium and grown in darkness for 64–68 h. Seedlings were placed into unilateral blue light and images were recorded every 10 min for 4 h with

a Retiga 6000 CCD camera (QImaging) connected to a personal computer running QCapture Pro 7 software (QImaging) with supplemental infra-red illumination. Hypocotyl curvature was measured from two biological replicates, with ~10 seedlings measured from each replicate, using Fiji software[63]. Circular histograms were produced using Oriana software (Kovach Computing Services). Box and whisker plots, and one-way analysis of variance (ANOVA) with Tukey post test, were performed using Graphpad Prism.

**Leaf positioning**. Seedlings were grown on soil for 9 d under 80 µmol m$^{-2}$ s$^{-1}$ white light before transfer to 10 µmol m$^{-2}$ s$^{-1}$ white light for 4 d. One cotyledon was removed, seedlings were placed flat on an agar plate, and plates were placed on a white light transilluminator and photographed. Petiole angles from the horizontal were measured from three biological replicates, with 20 seedlings for each replicate, using Fiji software. Box and whisker plots, and one-way ANOVA with Tukey post test, were performed using Graphpad Prism.

**Confocal microscopy**. Localisation of GFP-tagged NPH3 was visualised with a Leica SP8 laser scanning confocal microscope using HC PL APO ×20/0.75 or ×40/1.30 objective. The 488-nm excitation line was used, GFP fluorescence was collected between 500 and 530 nm and autofluorescence between 650 and 750 nm. Images were acquired at 1024–1024-pixel resolution with a line average of two. Z-stacks were acquired at 5 min intervals, with darkness between each scan. Maximum projection images were constructed from z-stacks using Fiji software.

**Immunoblot analysis**. Total proteins were extracted by grinding 50 3-d-old etiolated Arabidopsis seedlings or 1 cm Agrobacterium-infiltrated N. benthamiana leaf discs in 100 µl 2× sodium dodecyl sulfate (SDS) sample buffer under red safelight illumination, clarified by centrifugation at $13,000 \times g$ for 5 min, boiled for 4 min and subjected to SDS-PAGE. Proteins were transferred onto a polyvinylidene fluoride membrane (PVDF; Bio-Rad) with a Trans-Blot Turbo Transfer System (Bio-Rad) and detected with anti-GST monoclonal antibody (1:10,000, Merck 05-782), anti-HA monoclonal antibody (1:10,000, clone 3F10, Merck 11867423001) anti-GFP-HFP monoclonal antibody (1:5000, clone GG4-2C2.12.10, Miltenyi Biotech 130-091-833), anti-thiophosphoester monoclonal antibody (1:10,000, clone 51-8, Abcam ab133473), anti-UGPase antibody (1:20,000, AgriSera, AS05 086), anti-phot1 polyclonal antibodies[11] (1:10,000), anti-NPH3 purified polyclonal antibodies raised in rabbits against peptides IPNRKTLIEATPQSF and GVDHPPPRKPRRWRN (Eurogentec, 1:20,000) and polyclonal antibodies raised in rabbits against phosphorylated S744 of NPH3 using peptide KPRRWRNpSIS (where pS represents phosphorylated serine) as antigen (Eurogentec, 1:80,000). Blots were developed with horseradish peroxidase (HRP)-linked secondary antibodies (anti-rabbit IgG, Promega W4011; anti-mouse IgG, Promega W4021, rabbit anti-rat IgG, Dako P0450, donkey anti-goat IgG, Promega V8051), Immobilon Western Chemiluminescent HRP Substrate (Merck) and signals detected with a Fusion FX imaging system (Vilber) or by X-ray film (Carestream) and digitised with an Epson Perfection V700 flatbed scanner.

**Far-western blot analysis**. Arabidopsis 14-3-3 isoforms Epsilon and Lambda were expressed using the pGEX-4T1 vector (Merck), as a translational fusion with GST[32]. Recombinant 14-3-3 proteins were expressed using Escherichia coli BL21(DE3) (Merck) host strain, induced with 1 mM isopropyl β-D-1-thiogalactopyranoside at 37 °C for 2 h. Cells were lysed by sonication, purified with Pierce Glutathione Agarose resin (ThermoFisher) and eluted with 10 mM reduced glutathione in 50 mM Tris (pH 8.0), 150 mM NaCl. Total protein extracts were prepared from 3-d-old etiolated seedlings maintained in darkness, or following blue light irradiation, under a dim red safe light. Seedlings were ground in a mortar and pestle in GTEN buffer (10% [v/v] glycerol, 25 mM Tris-HCl [pH 7.5], 1 mM ethylenediaminetetraacetic acid (EDTA), 150 mM NaCl) supplemented with 0.5% SDS, 10 mM DTT, 1 mM phenylmethylsulfonyl fluoride (PMSF) and a protease inhibitor mixture (Complete EDTA-free; Merck) on ice and clarified by centrifugation at $10,000 \times g$, 4 °C for 10 min. IPs were performed using GFP-Trap Agarose beads (Chromotek), eluted by boiling in 2× SDS sample buffer, separated by SDS-PAGE and transferred to PVDF membrane. PVDF membranes were incubated with purified GST-14-3-3 proteins or GST alone in far-western buffer (20 mM HEPES-KOH [pH 7.7], 75 mM KCl, 0.1 mM EDTA, 1 mM DTT, 2% milk, 0.04% Tween-20) at a final concentration of 1 µM. 14-3-3 binding was detected using anti-GST monoclonal antibody (1:10,000, Merck 05-782).

**Immunoprecipitation**. Total protein extracts were prepared from 3-d-old etiolated WT seedlings or seedlings expressing GFP-NPH3 maintained in darkness (Dark) or irradiated with 20 µmol m$^{-2}$ s$^{-1}$ of blue light for 15 min. Seedlings were ground in a mortar and pestle in IP buffer (50 mM Tris-HCl [pH 7.5], 150 mM NaCl, 1 mM EDTA, 1% Triton X-100, 1 mM PMSF) supplemented with protease inhibitor mixture (Complete EDTA-free; Merck) and half-strength phosphatase inhibitor cocktail 2 and 3 (Merck). Samples were clarified twice by centrifugation at $14,000 \times g$, 4 °C for 10 min. IPs were performed using the µMACS GFP isolation kit (Miltenyi Biotech), eluted with 0.1 M Triethylamine pH 11.8/0.1% Triton X-100 and neutralised with 1 M MES pH 3. Proteins were identified by liquid chromatography-tandem MS using the Fingerprints Proteomics Facility

(University of Dundee). All proteins identified from two independent biological replicates are provided as Supplementary Data 1. Only proteins identified in two biological replicates from at least two peptides were retained for analysis. Proteins identified in IPs from WT seedlings were considered contaminants. Protein intensities were converted to the relative abundance of the bait protein (GFP-NPH3), which was set to 100 in each sample. Proteins showing at least a twofold change in relative abundance following blue light irradiation were identified (Supplementary Data 2).

**In vitro kinase assay**. The coding sequence of NPH3 in NPH3::GFP-NPH3 and NPH3::GFP-NPH3 S744A was amplified and inserted into the pSP64 poly(A) vector (Promega), together with an N-terminal Haemagglutinin (HA) tag, using Gibson Assembly (New England Biolabs). The RPT2, DOT3, NPY1 and NRL1-coding sequences were amplified from cDNA and inserted into the pSP64 poly(A) vector, together with an N-terminal GST tag, using Gibson Assembly (New England Biolabs). The RPT2 S591A substitution was introduced by site-directed mutagenesis. All primer sequences are available in Supplementary Table 1. In vitro kinase assays were performed by co-expressing the substrate together with a gatekeeper engineered phot1 (phot1$^{GK}$)[20] using the TnT® SP6 High-Yield Wheat Germ Protein Expression System (Promega). For each 20 µl cell-free expression reaction 2 µg of pSP64 poly(A) vector encoding phot1$^{GK}$ and 2 µg of the vector encoding the substrate were incubated in the presence of 10 µM FMN for 2 h, in darkness, at room temperature. Thiophosphorylation reactions were prepared under red safe light illumination. For each 20 µl reaction, 10 µl of cell-free expression sample was incubated in the presence of 500 µM N$^6$-benzyl-ATPγS (Jena Bioscience), in phosphorylation buffer contained 37.5 mM Tris-HCl pH 7.5, 5.3 mM MgSO4, 150 mM NaCl and 1 mM EGTA. Samples were either mock irradiated or treated for 20 s with white light at a total fluence of 60,000 µmol m$^{-2}$. Reactions were performed for 5 min and stopped by the addition of EDTA (pH 8.0) to a final concentration of 20 mM. Thiophosphorylated molecules were alkylated with 2.5 mM p-nitrobenzyl mesylate (Abcam) for 2 hours in darkness, at room temperature. HA-tagged NPH3 was immunoprecipitated using Pierce™ Anti-HA Magnetic Beads (Thermo Fisher Scientific). Thiophosphorylation was visualised by immunoblotting with anti-thiophosphoester monoclonal antibody (1:10,000, clone 51-8, Abcam ab133473).

**Reporting summary**. Further information on research design is available in the Nature Research Reporting Summary linked to this article.

## Data availability

The authors declare that all data presented in this study are available in the figures and the accompanying Supplementary Information file. MS proteomics data have been deposited to the ProteomeXchange Consortium via the PRIDE partner repository with the data set identifier PXD028698. Data that support the study are available from the corresponding authors upon reasonable request. Source data are provided with this paper.

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

## Acknowledgements

This work was supported by funding from the UK Biotechnology and Biological Sciences Research Council (BB/M002128/1, BB/R001499/1 to J.M.C.) and the Grant-in-Aid for Scientific Research Grant from the Japan Society for the Promotion of Science (15KK0254 to N.S.). We are grateful to Albertus H. de Boer for providing 14-3-3 quadruple mutant seed. We are indebted to Claudia Oecking for sharing data and helpful discussions.

## Author contributions

S.S, N.S. and J.M.C. designed research; S.S., T.W., L.H., D.P. and M.L. performed research; S.S, N.S. and J.M.C. analysed data; S.S. and J.M.C. wrote the manuscript. All authors commented on the manuscript.

## Competing interests

The authors declare no competing interests.
