## [Peer Review File · Nature Communications]

Regulation of Plant Phototropic Growth by NPH3/RPT2-like Substrate Phosphorylation and 14-3-3 BindingREVIEWER COMMENTS

Reviewer #1 (Remarks to the Author):

This is a very exciting paper for researchers who focus on phototropin signaling. Evidence that NPH3 is a phototropin1 target has been missing. It is not an intuitive fact, due to massive dephosphorylation events accompanied by light irradiation of NPH3. The direct link between phot1 signaling and NPH3 has been unidentified to date. Finally, the mechanism of phototropin action has been revealed and a whole new area of research depicting NRL proteins has opened.

The paper is very clearly written and easy to follow.

The molecular work presented in this paper is convincing. However, I would like to ask for more information or data on the physiological experiments. The authors need to clarify if they have assessed phototropism and leaf positioning in three independent experiments to obtain biological replicates. If not, I would like to ask for additional experiments and for the presentation of those results.

I would also like to ask the authors to make a comment on NPH3 interactions with Chi and Omega 14-3-3 proteins, which seem to be important interactor partners of NPH3 according to the MS analysis.

Minor remarks:

I would prefer if the authors use the whole phrase NPH3 S744A mutein, similarly for all other NPH3 variants.

Fig.2C. Two times M2 indicated

line 192. Based on the figure provided the S746A mutation prevented NPH3 relocalization upon light as well.

Fig.3C. I would argue, that phosphorylation at S746 contributes in a way to 14-3-3 binding.

Fig.4B. I assume that the higher molecular weight in transgenic plants of NPH3 as compared to the wild type in the NPH3 panel stems from the fusion with GFP. Please indicate this in the figure.

Fig.4C. Why is there a distortion of bands recognized by the pS744 antibody in the phot2 mutant? Is there a difference in the dephosphorylation pattern between LB and HB in the phot2 mutant?

Fig.6., Fig.S2., Fig.S3C. Please indicate the *nph3* mutant using italics, please provide data from 3 biological replicates. Please add statistical analysis at a chosen time point for hypocotyl curvatures.

Fig. 7F, G., Fig. S3A, B. Please add control lines on those graphs. I assume that the experiment has been performed simultaneously with those in Fig.6. I also think that the scale bar has been adjusted to ease comparisons, however, I would prefer to have control lines on one graph with the tested lines. Please indicate the light intensity on graphs as in Fig.6.

Methods:

Please indicate the catalog number of anti-UGPase antibodies from Agrisera.

Please add a list of used primers.

Please indicate information about the cloning of other NRL proteins e.g. DOT3, NPY1, NRL1, RPT2.

Reviewer #2 (Remarks to the Author):

In this manuscript, the authors present highly original and extremely noteworthy results. They perform MS analysis to identify binding partners of NPH3, a key regulator of phototropism. They find that 14-3-3 proteins bind to NPH3, specifically to a phosphorylated C-terminal serine residue. The serine is rapidly phosphorylated by phototropin 1 in response to blue light. This phosphorylation of NPH3 by phot1 is the 'missing link' of phototropism research and represents a big step in our understanding of how plant growth is directed towards light. The MS is already looking great, but as a reviewer it is my duty to point out areas that feel could be further improved and I outline these points below.

The methodology in the paper is innovative and yet sound and data analysis is of a level expected for the field. In the methods section, several techniques are cited, rather than described. To ensure reproducibility, please describe these methods. Some continuous data are presented as histograms, where violin/ box plots would give the reader a better understanding of the variation in these experiments.

I find most of the author's conclusions to be reasonable, but in one case a lack of controls limits the strength of the conclusion. In Figure 7G/7F and S3A/B, there is no WT or *nph3* control plants. Without these controls it is not possible to say that phototropism is reduced in these mutants (results from this experiment could alternatively be confounded by malfunctioning light source etc.)

I would also like for the authors to clarify the statements in line 704. On first reading I thought that the authors were suggesting that residual phototropism in the *nph3*/ GFP-NPH3-S744A mutant was due to redundancy with other NRLs. But if that is the case, residual function would also be seen in the *nph3* mutant. Do the authors mean that GFP-NPH3-S744A has some effect on the native NRLs? If there is no evidence of NPH3 action through other NRLs, I would say that the most simple answer is that GFP-NPH3-S744A retains some residual functionality (perhaps through phosphorylation of S746?). A minor role for phosphorylation of S746 is suggested by the slightly reduced function of GFP-NPH3-S746A compared to GFP-NPH3 (Figure 6B & 6E).

In addition to these comments, I also have some minor points that I feel would enhance the readability of this MS:

Line 52. Syntax: is “although” appropriate in this sentence?

Figure 2C. (presumably mislabelled M2 = M3?). The authors could consider adding a line in the text to acknowledge that it was unexpected that the M3 runs at the same speed as the phosphorylated form of GFP-NPH3. Later in the MS this is explained somewhat, but it could be clearer.

The authors often refer to the faster running NPH3 as completely de-phosphorylated, but this sometimes leads to confusing sentences. For example “Complete dephosphorylation of NPH3 required 15 min of blue light irradiation (Fig. 5A), whereas phosphorylation of S744 was detected within 30 s and maintained over the 2 h irradiation period”. How can NPH3 be 'completely dephosphorylated' and phosphorylated at the same time? Referring to the band shift instead could help to resolve this, e.g. “Enhanced mobility of NPH3 required 15 min of blue light irradiation (Fig. 5A), whereas phosphorylation of S744 was detected within 30 s and maintained over the 2 h irradiation period”

Consider swapping the numbering of Figure 7 so that 7F is discussed in the text before 7G

More information into the figure legends would improve readability. For example, it's not immediately clear whether transgenes are being expressed in the WT or *nph3* background. This is important for the interpretation of these results and so it would be useful to state it in the figure/ figure legend.

Best wishes,

Scott Hayes

We would like to thank the reviewers for their time and suggestions on how to improve our manuscript. We have addressed each of the comments below.

Reviewer #1 (Remarks to the Author):

This is a very exciting paper for researchers who focus on phototropin signaling. Evidence that NPH3 is a phototropin1 target has been missing. It is not an intuitive fact, due to massive dephosphorylation events accompanied by light irradiation of NPH3. The direct link between phot1 signaling and NPH3 has been unidentified to date. Finally, the mechanism of phototropin action has been revealed and a whole new area of research depicting NRL proteins has opened. The paper is very clearly written and easy to follow.

We are grateful to the reviewer for their positive comments on our manuscript and that they share our excitement in these results.

The molecular work presented in this paper is convincing. However, I would like to ask for more information or data on the physiological experiments. The authors need to clarify if they have assessed phototropism and leaf positioning in three independent experiments to obtain biological replicates. If not, I would like to ask for additional experiments and for the presentation of those results.

The phototropism data is from two biological replicates, with approximately 10 seedlings measured for each replicate. Due to the time-consuming nature of these experiments to perform and analyse (the data presented represents 376 h of time-lapse imaging of over 900 seedlings, with over 23,000 measurements), we believe that two biological replicates are a reasonable compromise that still allows us to make robust conclusions. We have also provided datasets from two independent transgenic lines, and performed phototropism under three different fluence rates, with consistent results. Furthermore, we observe comparable levels of reduced functionality for all transgenic lines unable to bind 14-3-3 proteins (GFP-NPH3-S744A, GFP-NPH3-S744A S746A, GFP-NPH3-S744D S746D and GFP-NPH3-mR18) and comparable levels of functionality for all transgenic lines that retain light-dependent 14-3-3 binding (GFP-NPH3 and GFP-NPH3-S746A). For the petiole positioning, we have performed additional experiments and the data presented in Fig. 6E and Fig. S3 is now from three biological replicates. Details of the number of replicates, and number of measurements per replicate, have been added to the Methods section.

Would also like to ask the authors to make a comment on NPH3 interactions with Chi and Omega 14-3-3 proteins, which seem to be important interactor partners of NPH3 according to the MS analysis.

While Chi and Omega are the two 14-3-3 isoforms identified with the highest relative abundance in our MS analysis (having 15.9 % and 9.3 % respectively), there are no significant differences between the amounts of any of the 14-3-3 isoforms in the Light samples (assessed by one-way ANOVA) and therefore we would be reluctant to draw any conclusions regarding which isoforms bind most to phosphorylated NPH3. The consistent difference is the (on average) 20-fold increase in abundance observed for each 14-3-3 isoform in the Light samples compared to the Dark samples.

I would prefer if the authors use the whole phrase NPH3 S744A mutein, similarly for all other NPH3 variants.

Thank you for this suggestion, we have made these changes throughout the manuscript.

Fig.2C. Two times M2 indicated.

We are sorry for the mistake in Fig. 2C and have corrected this.

line 192. Based on the figure provided the S746A mutation prevented NPH3 relocalization upon light as well.

While mutation of S744 prevented any detectable relocalisation, the NPH3 S746A mutant did show a very slight response. I appreciate that it is probably only visible if you zoom in on the image, but there is a difference between the dark and light images.

Fig.3C. I would argue, that phosphorylation at S746 contributes in a way to 14-3-3 binding.

We agree that the NPH4 S746A mutant is affected in 14-3-3 binding and on lines 220-222 state:

“Binding of recombinant 14-3-3 Epsilon was evident for both GFP-NPH3 and NPH3 S746A mutant in a light-dependent manner, with the signal for S746A being substantially lower.”

Similarly, we also observed reduced binding of the S744 phospho-specific antibody to NPH3 S746A and on lines 220-222 state:

“Phosphorylation of S744 was also detectable for NPH3 S746A mutant expressing seedlings at a reduced level, similar to the results observed for 14-3-3 binding (Fig. 3C).”

However, we observe no functional consequence of the S746 mutation *in planta* and therefore any reduction is insufficient to impact signalling.

Fig.4B. I assume that the higher molecular weight in transgenic plants of NPH3 as compared to the wild type in the NPH3 panel stems from the fusion with GFP. Please indicate this in the figure.

Yes, the difference in molecular weight is due to the presence of the GFP tag fused to NPH3 in the transgenic lines. We have now indicated this in the Fig 4B.

Fig.4C. Why is there a distortion of bands recognized by the pS744 antibody in the *phot2* mutant? Is there a difference in the dephosphorylation pattern between LB and HB in the *phot2* mutant?

There is no difference in NPH3 dephosphorylation between LB and HB in the *phot2* mutant, as can be seen from the anti-NPH3 blots in the second panel of Fig. 4C. The distortion of the bands seen with the anti-phosphoS744 antibody in the upper panel, indicating phosphorylation of S744 in both LB and HB conditions in *phot2* seedlings, are also evident in the lower anti-UGPase panel, and are due to uneven migration of the samples during SDS-PAGE.

Fig.6., Fig.S2., Fig.S3C. Please indicate the *nph3* mutant using italics, please provide data from 3 biological replicates. Please add statistical analysis at a chosen time point for hypocotyl curvatures.

We have indicated the *nph3* mutant with italics for these figures.

We have performed statistical analysis for the phototropism data using the final angle of curvature from the 240 min time-point. This data is presented in the form of box and whisker plots with one-way ANOVA analysis in Fig. S2 and Fig. S4.

Fig. 7F, G., Fig. S3A, B. Please add control lines on those graphs. I assume that the experiment has been performed simultaneously with those in Fig.6. I also think that the scale bar has been adjusted to ease comparisons, however, I would prefer to have control lines on one graph with the tested lines. Please indicate the light intensity on graphs as in Fig.6.

We have added control line data for these experiments and re-plotted the data so that the results for each fluence rate are presented on a single graph, as in Fig. 6. Therefore, the data from Fig. S3A and B is now incorporated into Fig 7F, G and H. We hope these changes make the data clearer. Light intensity is now indicated on each graph.

Please indicate the catalog number of anti-UGPase antibodies from Agrisera.

This has been added.

Please add a list of used primers.

A list of primers is now included as Table S2.

Please indicate information about the cloning of other NRL proteins e.g. DOT3, NPY1, NRL1, RPT2.

This has been added to the Methods section and the primers included in Table S2.

Reviewer #2 (Remarks to the Author):

In this manuscript, the authors present highly original and extremely noteworthy results. They perform MS analysis to identify binding partners of NPH3, a key regulator of phototropism. They find that 14-3-3 proteins bind to NPH3, specifically to a phosphorylated C-terminal serine residue. The serine is rapidly phosphorylated by phototropin 1 in response to blue light. This phosphorylation of NPH3 by phot1 is the 'missing link' of phototropism research and represents a big step in our understanding of how plant growth is directed towards light. The MS is already looking great, but as a reviewer it is my duty to point out areas that feel could be further improved and I outline these points below.

We are grateful to the reviewer for their positive comments and hope they find our revisions have improved the manuscript.

The methodology in the paper is innovative and yet sound and data analysis is of a level expected for the field. In the methods section, several techniques are cited, rather than described. To ensure reproducibility, please describe these methods.

Details of cited methods have now been added.

Some continuous data are presented as histograms, where violin/ box plots would give the reader a better understanding of the variation in these experiments.

Thank you for this suggestion, we now present the petiole positioning data in Fig. 6E and Fig. S3 as box and whisker plots. We have also added statistical analysis for the phototropism data, as suggested by Reviewer #1, and present these data as box and whisker plots in Fig. S2 and Fig. S4.

I find most of the author's conclusions to be reasonable, but in one case a lack of controls limits the strength of the conclusion. In Figure 7G/7F and S3A/B, there is no WT or *nph3* control plants. Without these controls it is not possible to say that phototropism is reduced in these mutants (results from this experiment could alternatively be confounded by malfunctioning light source etc.)

We have added control line data for these experiments and re-plotted the data so that the results for each fluence rate are presented on a single graph, therefore the data from Fig. S3A and B is now incorporated into Fig 7F, G and H.

I would also like for the authors to clarify the statements in line 704. On first reading I thought that the authors were suggesting that residual phototropism in the *nph3*/ GFP-NPH3-S744A mutant was due to redundancy with other NRLs. But if that is the case, residual function would also be seen in the *nph3* mutant. Do the authors mean that GFP-NPH3-S744A has some effect on the native NRLs? If there is no evidence of NPH3 action through other NRLs, I would say that the most simple answer is that GFP-NPH3-S744A retains some residual

functionality (perhaps through phosphorylation of S746?). A minor role for phosphorylation of S746 is suggested by the slightly reduced function of GFP-NPH3-S746A compared to GFP-NPH3 (Figure 6B & 6E).

Yes, we are suggesting that the residual phototropism in the GFP-NPH3-S744A mutant could arise from redundancy with other NRLs. It is possible that phosphorylation of other NRLs by phot1 can support signalling in the presence of a non-phosphorylatable NPH3 mutant, but not in the complete absence of NPH3, as in the *nph3* mutant. For example, NPH3 could be required for mediating protein-protein interactions within a signalling complex. However, this is speculation as we do not have evidence of co-action with other NRLs. We agree that we cannot exclude that the GFP-NPH3-S744A mutant retains residual functionality, although this raises the question of how a directional response is instigated. The possibility of a gradient of phosphorylation of other NRL family members would provide an answer to this. We do not believe the data supports that the GFP-NPH3-S746A mutant has reduced functionality, as there are no significant differences observed compared to GFP-NPH3 for phototropism (Fig.S2) or petiole positioning (Fig. 6E).

Line 52. Syntax: is "although" appropriate in this sentence?

Thank you for pointing out this mistake, we have modified this sentence as follows:

"Although multiple phosphorylation sites have been identified within phot1 and phot2¹², only sites within the kinase activation loop have been shown to be important for signalling"

Figure 2C. (presumably mislabelled M2 = M3?). The authors could consider adding a line in the text to acknowledge that it was unexpected that the M3 runs at the same speed as the phosphorylated form of GFP-NPH3. Later in the MS this is explained somewhat, but it could be clearer.

Yes, we are sorry for the mistake in Fig. 2C and have corrected this. We have modified the text on lines 185-190 to try and make this clearer:

"Therefore, mutation of phosphorylation sites within the M3 region at the C-terminus of NPH3 prevents both blue light-induced dephosphorylation of sites within the NPH3 domain, which contribute to reducing the electrophoretic mobility, as well as subcellular relocalisation into aggregates."

The authors often refer to the faster running NPH3 as completely de-phosphorylated, but this sometimes leads to confusing sentences. For example "Complete dephosphorylation of NPH3 required 15 min of blue light irradiation (Fig. 5A), whereas phosphorylation of S744 was detected within 30 s and maintained over the 2 h irradiation period". How can NPH3 be 'completely dephosphorylated' and phosphorylated at the same time? Referring to the band shift instead could help to resolve this, e.g. "Enhanced mobility of NPH3 required 15 min of blue light irradiation (Fig. 5A), whereas phosphorylation of S744 was detected within 30 s and maintained over the 2 h irradiation period".

We are sorry for any confusion and have modified the text throughout the manuscript to clarify the differences between the phosphorylation status of sites within the NPH3 domain, which contribute to changes in electrophoretic mobility to distinguish this from changes in phosphorylation at S744. For example, lines 264-267 now read:

"Dephosphorylation of sites which alter NPH3 electrophoretic mobility required 15 min of blue light irradiation (Fig. 5A), whereas phosphorylation of S744 was detected within 30 s and maintained over the 2 h irradiation period."

Consider swapping the numbering of Figure 7 so that 7F is discussed in the text before 7G.

We have changed the presentation of this data, so the data for each fluence rate is plotted on a single graph, which avoids this problem.

More information into the figure legends would improve readability. For example, it's not immediately clear whether transgenes are being expressed in the WT or *nph3* background. This is important for the interpretation of these results and so it would be useful to state it in the figure/ figure legend.

We have altered the figure legends to make it clear that all the transgenics are in the *nph3* mutant background.

REVIEWERS' COMMENTS

Reviewer #1 (Remarks to the Author):

I feel satisfied with all the changes in the manuscript and answers to my questions prepared by the Authors. I recommend this paper for publication in Nature Communications.

Reviewer #2 (Remarks to the Author):

All of my comments about the previous manuscript have been fully addressed.

Congratulations to all the authors, fantastic work!

Best wishes,

Scott Hayes